# Automatic Classification Framework of Tongue Feature Based on Convolutional Neural Networks

**DOI:** 10.3390/mi13040501

**Published:** 2022-03-24

**Authors:** Jiawei Li, Zhidong Zhang, Xiaolong Zhu, Yunlong Zhao, Yuhang Ma, Junbin Zang, Bo Li, Xiyuan Cao, Chenyang Xue

**Affiliations:** Key Laboratory of Instrumentation Science & Dynamic Measurement, North University of China, Taiyuan 030051, China; jiaweili0123@163.com (J.L.); zxldtc112233@163.com (X.Z.); mailzyl@163.com (Y.Z.); myh_694@163.com (Y.M.); zangjunbin@nuc.edu.cn (J.Z.); lb@nuc.edu.cn (B.L.); caoxiyuan@nuc.edu.cn (X.C.); xuechenyang@nuc.edu.cn (C.X.)

**Keywords:** TCM tongue diagnosis, deep learning, convolutional neural network, tongue segmentation, image classification

## Abstract

Tongue diagnosis is an important part of the diagnostic process in traditional Chinese medicine (TCM). It primarily relies on the expertise and experience of TCM practitioners in identifying tongue features, which are subjective and unstable. We proposed a tongue feature classification framework based on convolutional neural networks to reduce the differences in diagnoses among TCM practitioners. Initially, we used our self-designed instrument to capture 482 tongue photos and created 11 data sets based on different features. Then, the tongue segmentation task was completed using an upgraded facial landmark detection method and UNET. Finally, we used ResNet34 as the backbone to extract features from the tongue photos and classify them. Experimental results show that our framework has excellent results with an overall accuracy of over 86 percent and is particularly sensitive to the corresponding feature regions, and thus it could assist TCM practitioners in making more accurate diagnoses.

## 1. Introduction

Traditional Chinese medicine (TCM) has received increasing attention and acknowledgment from medical experts since the World Health Organization (WHO) included it in the latest global medical guidelines [1,2]. Tongue diagnosis is a key criterion for TCM diagnosis; it is a noninvasive and convenient approach to assess human health [3,4]. However, tongue features are varied and intertwined, and subtle differences in features may correspond to completely different diseases. TCM practitioners rely on their expertise and previous experience to identify tongue features, which are susceptible to environmental factors, resulting in unstable and inaccurate diagnoses. Therefore, building a system that can objectively classify tongue features is urgently required.

Generally, tongue feature classification refers to identifying the types of different tongue features and assisting TCM practitioners to confirm diagnoses. Accurate classification is difficult due to the minor differences within the inner class. In addition, this classification mainly extracts and classifies the feature vector of the tongue body images, rather than the entire tongue regions. However, the tongue image captured by the camera has other information including lips, teeth, and even facial images. Image segmentation should be used to eliminate useless information, which can increase the accuracy of the classification and reduce the amount of calculation. Therefore, objectification and automated tongue diagnosis are mainly composed of two tasks; tongue image segmentation and tongue feature classification. Segmenting the tongue from the image is a prerequisite for tongue feature classification [5].

For tongue segmentation, traditional methods are generally used, including the region growing method, threshold method, watershed transformation method, edge detection, and snake model [6,7,8,9,10,11]. Nevertheless, the color around the tongue is similar to the color of the tongue’s body, and the edge contour is relatively fuzzy. The application of traditional methods to achieve the effect of tongue image segmentation is not ideal. In recent years, more studies on using semantic segmentation based on deep convolutional neural networks (CNN) for tongue segmentation have been conducted, and the effect is better than some traditional image segmentation methods [12,13]. However, most training data sets used only contain normal tongue images, thereby reducing the clinical practicability and robustness of the segmentation algorithm.

In the research of tongue classification, many scholars digitize tongue features and classify them according to particular threshold standards [14,15]. Nevertheless, the accuracy of recognition is unsatisfactory. A number of scholars have also developed classification models for various tongue features using CNN. They use convolution operations to extract high-level semantic features and support vector machine (SVM), softmax, or other classifiers for classification. These approaches have been shown to be effective in the recognition of various features of tongue texture, tongue coating, and tongue shape [3,16,17,18]. These studies, however, only focus on a few features and do not systematically obtain all tongue features for analysis based on the theory of TCM tongue diagnosis; thus, the results lack clinical significance.

In this paper, we use the CNN approach to deal with these limitations. The main contribution and innovation of this paper is the optimization of the usability, efficiency, and interpretability of the tongue diagnosis classification method. First, our method is to classify 11 tongue features according to TCM theory, which can identify more comprehensive tongue features and is consistent with TCM theory to assist practitioners in making more standard diagnoses. Therefore, a clinical and standard data set with 11 tongue features was constructed due to the lack of a relevant open-source data set. In contrast, most previous studies have classified one or a few features, which are not applicable to clinical use. Second, we propose a tongue region extraction method, which can convert the original image of the tongue into a smaller pixel image before segmentation and classification to improve the efficiency and accuracy of segmentation and classification. Finally, we use GradCAM to visualize the decisions of the classification network, enabling the user to know where anomalous features appear and to better understand the basis of the classification. The proposed method is described in detail in Section 2, and the experimental results are presented in Section 3. Section 4 concludes this study.

## 2. Related Works

The main challenge in the domain of tongue diagnosis classification systems is to digitize the diagnostic experience of TCM practitioners in order to allow models to make as clinically meaningful and accurate diagnoses as TCM practitioners do. Such models are generally composed of two parts: tongue segmentation and classification of tongue features.

In the field of tongue segmentation, Li et al. [19] transformed the tongue image into HSI color space and used thresholding to obtain the initial region of the tongue body, and then used the tongue root and upper lip gap region to remove the irrelevant region to achieve the final segmentation. Shi et al. [20] combined the geometrical Snake model with the parameterized GVFSnake model to establish the C2G2FSnake tongue segmentation model. They introduced color space information as control information to update the external force parameters to control the accuracy and velocity of the curve, and then applied the gradient vector flow field (GVF) to obtain the results. However, these methods based on traditional image processing techniques will occasionally judge similarly colored regions around the tongue as tongue pixels as well, and the segmentation edge is rough. The segmentation effect of the traditional method is not ideal. In [12,13,21,22], semantic segmentation methods such as FCN, SegNet, and ResNet were applied to tongue segmentation. These methods using CNN for feature extraction can precisely distinguish the difference between tongue pixels and irrelevant pixels, and the contour of the segmentation result is smooth. Also, these methods do not require manual assistance at all. However, the data sets referenced by these methods are normal tongue images and lack clinicality. Therefore, this paper uses semantic segmentation based on CNN for the tongue segmentation and uses clinical tongue images as the data set for training, which is more clinically relevant.

In the field of tongue feature classification, Zhang et al. [14] extracted the tongue measurements, distances, areas, and their proportions from the tongue image and classified the tongue shape using a decision tree. Bo et al. [15] corrected the tongue deflection by applying three geometric criteria and then classified tongue shapes by analytic hierarchy process according to seven geometric features defined by various measurements of length, area, and angle of the tongue. These methods are used to convert specific tongue features into digital quantities by corresponding image processing, and then classifiers such as decision tree or SVM are used to achieve feature classification. However, the classification accuracy of these methods is not ideal, and it can only classify a few specific features, which is not very practical. Li et al. [3] proposed a method for the recognition of tooth marks. The method first generates suspicious regions based on concavity information, then extracts the depth features within the regions by CNN, and finally uses multiple-instance SVM to complete the final classification. Tang et al. [16] selected suspected rotten-greasy tongue coating patches firstly, and then used ResNet to extract features of each patch to complete the classification of the curdy or greasy coating by MI-SVM. These tongue feature classification methods based on deep learning can automatically extract more detailed and relevant features by CNN for different tongue features with higher accuracy. However, these classification methods currently classify only a few tongue features and cannot effectively assist physicians in clinical diagnosis. In addition, interpretability is very important for medical diagnosis results, but these methods are to draw direct conclusions without explainable process. Therefore, we built a classification model for 11 tongue features using CNN and visualized the decisions for the classification using GradCAM.

## 3. Methods

A tongue feature classification method is proposed, and the entire framework detail is shown in Figure 1. To extract the tongue region from the original input images, the recognition algorithm with 68 facial landmarks in the dlib library is upgraded. Then, the tongue images are segmented by the UNET [23]. Lastly, we build 11 tongue classification models by using the Residual Network (ResNet) [24] according to the different features. In addition, the Grad-CAM [25] is used to provide visual explanations for model decisions.

### 3.1. Data Sets

The tongue images in the data set were collected with consistent conditions and high resolution to ensure the objectivity and effectiveness of the classification. The tongue images of the data set, which contains 482 photos with the pixel size of 3264 × 2448, were collected by a custom designed tongue diagnosis instrument (Figure 2). The tongue image was then classified by the expert TCM practitioners from Shanxi University of Chinese Medicine’s Affiliated Hospital who have received extensive training and have normal or corrected vision. Eleven datasets were created based on different features, including six tongue body features and five tongue coating features (Table 1).

### 3.2. Extraction of the Tongue Region

The original image captured by the tongue diagnostic instrument is a photograph containing the entire face as well as the edges of the instrument. The extraction of the tongue region can improve the efficiency of tongue segmentation by accurately labeling the tongue segmentation and reducing the amount of image segmentation computation. However, no detector for images of sticking out tongues is available at present. In this stage, we upgraded the facial landmark recognition algorithm in the dlib library to extract the tongue region. The steps were as follows:

Step 1: Load the original image into the facial detector and obtain a matrix containing pixel coordinate information of all 68 facial landmarks. Extract the coordinates of each landmark using traversal and label them with sequential numbers. Then, record the coordinates of the 49th, 55th, and 9th of the 68 landmarks, which are the landmarks on the sides of the mouth and the bottom of the chin respectively.

Step 2: Calculate the bounding box of the tongue area according to the coordinate difference ratio in the result of step 1. The height, width, and center point coordinates of the bounding box are calculated as follows:(1)$$\mathrm{height}={\;y}_{3}-\frac{y_{1}+y_{2}}{2}+\frac{x_{2}-x_{1}}{2}$$
(2)$$\mathrm{width}=2\times \left(x_{2}-x_{1}\right)$$
(3)$$\left[\begin{array}{c} x_{\mathrm{center}}\\ y_{\mathrm{center}} \end{array}\right]=\left[\begin{array}{c} \frac{x_{1}+x_{2}}{2}\\ y_{3}+\frac{x_{2}-x_{1}}{4}-\frac{\mathrm{height}}{2} \end{array}\right]$$
where (x_1_, y_1_) and (x_2_, y_2_) are the coordinates of the landmarks on the sides of the mouth, and (x_3_, y_3_) are the coordinates of the landmark on the bottom of the chin.

Step 3: Cut out the tongue region from the original image (Figure 3b) according to the bounding box.

We also built a data set for tongue segmentation after completing the task of extracting the tongue region of the original images. We manually labeled the edges of the tongue with the labelme software (Figure 3c) and generated corresponding contour images (Figure 3d). Tongue feature annotations were included with each contour image, and the data set was divided into the training set, validation set, and test set for the training of the segmentation model.

### 3.3. Segmentation of Tongue Images

As stated in Section 1, tongue segmentation can improve the effect of tongue feature classification by eliminating invalid information from the images. In this stage, we selected the UNET [23] based on deep CNN to complete the task of tongue segmentation. This method judges the category of each pixel in the image to obtain the tongue contour.

Structure: The structure of UNET can be divided into two parts (Figure 4). The first half of the UNET is the backbone feature extraction network. VGG16 [26] was chosen for feature extraction, which is a stack of convolution and maximum pooling operations. A feature layer with a new scale can be acquired after each pooling, resulting in five feature layers with distinct scales. The up-sampling part takes up the second half. The five feature layers were merged through the up-convolution method to produce an effective feature layer that contains all the features. The category of each pixel can be predicted according to the last obtained effective feature layer.Training: The data set for tongue segmentation was divided into the training set, validation set, and test set according to 8:1:1. The data were enhanced prior to training by random rotation and horizontal flipping of the image, as well as normalization. The loss included cross-entropy loss and dice loss. Adam algorithm was applied for optimization. Then, we used the official weight of the UNET network in the ImageNet data set as the initial weight for transfer learning. A total of 160 rounds were used to train the network. The weights of the backbone network were frozen in the first 80 rounds for rough training, and the learning rate was 1 × 10^−4^. The global network was trained with a learning rate of 1 × 10^−5^ in the last 80 rounds for fine training.Image Processing: The segmented contour images were processed by grayscale. Through observation, the generated gray image had a single gray level, with black pixels in the outer circle. Therefore, the grayscale image could be used as a mask to perform AND operate on the original tongue region image to realize the separation of the tongue. Then, we appended corresponding labels to the segmented tongue images to create classification data sets, which were also divided into the training, validation, and test sets for the training of the classification models.

### 3.4. Classification

We used the ResNet-34 [24] networks to classify tongue features.

Structure: Residual Network (ResNet-34) is a deep CNN with 34 layers, including 16 residual blocks, each with two layers (Figure 5). The last layer is an FC layer for tongue feature classification. The residual network increases the depth of the network through the connection of multiple residual blocks, while also avoiding the problem of gradient disappearance or gradient explosion.Training: The data sets for tongue feature classification are divided into the training set, validation set, and test set according to 6:2:2. The data are enhanced and normalized in the same way as the segmentation network preprocessing. The official model trained by ImageNet is used for initialization. The training is terminated after 160 rounds at a learning rate of 1 × 10^−4^. The models are trained separately for 11 different tongue feature data sets. At the end of the network, the output layer is adjusted accordingly to the number of internal categories of the different features, and the final classification judgment is made using the argmax function.

### 3.5. Feature Visualization

In this stage, we used Grad-CAM [21] to produce visual explanations for decisions from tongue feature classification models.

TCM practitioners perform tongue diagnoses by identifying corresponding features in various positions on the tongue. Therefore, whether the corresponding feature area contributes more in tongue feature classification models should be determined. The GradCAM algorithm is a method for visualizing the feature maps in CNNs. The algorithm assigns importance values to each neuron through the gradient information of the back propagation of the network, and then generates a heat map by linear weighted summation to obtain the regions that the model focuses on. The tongue feature classification model can be divided into two parts: feature extraction and classifier. The classifier only classifies based on the extracted features, while the last convolutional layer of the feature extraction part is the one with the richest semantic information of tongue features. Therefore, the feature map of this layer is used for visualization. The specific realization of visualization of the corresponding regions of tongue features is as follows:

Step 1: Calculating the gradient of the classification score of the tongue feature to the last convolutional layer in conv5_x of ResNet-34. Then, similar to global average pooling, each pixel value is averaged in each channel dimension to obtain the neuron importance weight. The formula is as follows:(4)$$\alpha_{k}^{c}=\frac{1}{Z}\underset{i}{\sum }\underset{j}{\sum }\frac{\partial y^{c}}{\partial A_{\mathrm{ij}}^{k}}$$
where Z represents the number of pixels in the feature map, and $A_{\mathrm{ij}}^{k}$ represents the pixel value at position (i, j) of the k-th feature map.

Step 2: the neuron importance weight is multiplied by each channel, added, and finally rectified by ReLU to obtain the heat map. The formula is as follows:(5)$$L_{\mathrm{Grad}-\mathrm{CAM}}^{c}=\;\mathrm{ReLU}\left(\underset{k}{\sum }\alpha_{k}^{c}A^{k}\right)$$

## 4. Results

In this section, we present the experimental results of the proposed method. The metrics pixel accuracy (PA) and mean intersection over union (MIoU) are used to evaluate the effect of tongue image segmentation, and the accuracy (Acc) and F1-Score are adopted to evaluate the final classification results. The calculation formulas of PA, MIoU, Acc, and F1-Score are as follows:(6)$$\mathrm{PA}=\frac{\sum_{i=0}^{k}p_{\mathrm{ii}}}{\sum_{i=0}^{k}\sum_{j=0}^{k}p_{\mathrm{ij}}}$$
(7)$$\mathrm{MIoU}=\frac{1}{k+1}\overset{k}{\underset{i=0}{\sum }}\frac{p_{\mathrm{ii}}}{\sum_{j=0}^{k}p_{\mathrm{ij}}+\sum_{j=0}^{k}p_{\mathrm{ji}}-p_{\mathrm{ii}}}$$
(8)$$\mathrm{Acc}=\frac{\mathrm{TP}+\mathrm{TN}}{\mathrm{TP}+\mathrm{FN}+\mathrm{FP}+\mathrm{TN}}$$
(9)$$F1-\mathrm{score}=\frac{1}{c}\overset{c}{\underset{k=1}{\sum }}\frac{2p_{k}\times r_{k}}{p_{k}+r_{k}}$$
where k represents the number of categories other than the background, and $p_{\mathrm{ij}}$ represents the number of pixels belonging to the i category, predicted to be the j category pixels. Therefore, $p_{\mathrm{ii}}$ represents true positive (TP), and $p_{\mathrm{ij}}$ and $p_{\mathrm{ji}}$ are false positive (FP) and false negative (FN) respectively; $p_{k}$ represents precision, and $r_{k}$ represents recall rate.

### 4.1. Results of Tongue Image Segmentation

We have compared the GrabCut [27] algorithm, which is based on edge detection of image processing, with the UNET segmentation model after the same processing to verify the feasibility of the segmentation model. The effect comparison is shown in Figure 6, and the values of PA and MIoU corresponding to the segmentation results are shown in Table 2. The GrabCut algorithm has more isolated noises in the segmentation. On the contrary, the UNET algorithm based on deep CNN has a better tongue image segmentation effect.

### 4.2. Results of Tongue Feature Classification

To verify the feasibility of the classification models, we test them on the test data sets. Figure 7 visualizes the result of tongue image classification. Table 3 shows the specific result parameters of the models for identifying the different categories of tongue features. The data show that the recognition effect is ideal, with an overall recognition accuracy of 86.14% and an overall F1-Score of 80.06%.

### 4.3. Visualization of the Indicator Regions of Tongue Feature Classification

The Grad-CAM algorithm is used to process the final convolution layer of the classification model. Figure 8 shows the results of part of the feature classification model. The red area in the heat map has a greater impact on recognition, whereas the blue area has less impact. The observation shows that the classification model is more sensitive to the corresponding characteristic regions, confirming that it has high practical clinical value.

## 5. Conclusions

We have proposed a tongue feature classification method based on CNN. The proposed method has three stages. First, the upgraded dlib facial 68 landmark algorithm is used to extract the tongue region. Second, the UNET network is used to accurately segment the tongue image. Finally, the ResNet network is used as the backbone feature extraction network to achieve the final classification of the characteristics of tongue texture and tongue coating and visualize the decisions using GradCAM. The experiment shows that the method has good accuracy for tongue feature recognition, with an overall accuracy rate of over 86%, and it is more sensitive to the corresponding feature regions. Compared with other approaches, our method identifies more comprehensive features and is more in line with TCM theory, which is suitable for clinical tongue diagnosis. When patients register at the hospital or have remote treatment, the approach can complete the classification of tongue features in advance, saving the time of TCM practitioners and thus allowing more patients to receive treatment. In addition, the GradCAM algorithm is applied to obtain a visible reference position of the network’s decisions, which can help users to know the location of their tongue abnormalities and help novice practitioners to gain experience. However, the approach is not applicable to children with small faces or those who have difficulty extending their tongues because the tongue diagnostic instrument cannot perform accurate tongue acquisition in these individuals.

In addition, the proposed method can be found to have some deviations in the evaluation indexes when performing the classification of tongue color features. We believe there are two reasons for the phenomenon. One is because there is little variation in the inter-class features of tongue color, the confidence levels of these classes are similar and not high enough in the final fully connected layer of the network. The method selects the class with the highest confidence level by the argmax function, which leads to insufficient accuracy. Another is that the number of samples within the class of the data set for tongue color features is unevenly distributed, making the classification of these features less accurate. Therefore, the method still needs to be improved for the network model and data sets afterwards.

## Figures and Tables

**Figure 1 micromachines-13-00501-f001:**
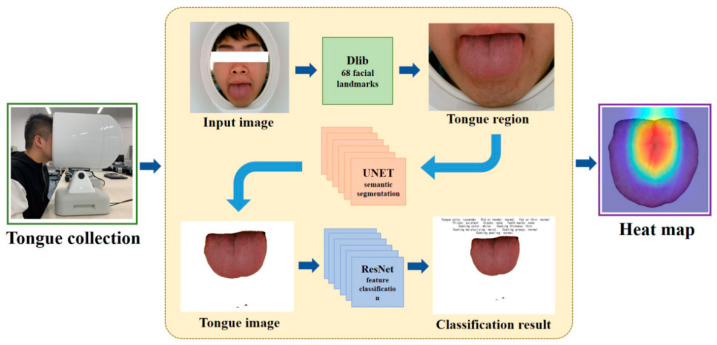
Overview of our framework.

**Figure 2 micromachines-13-00501-f002:**
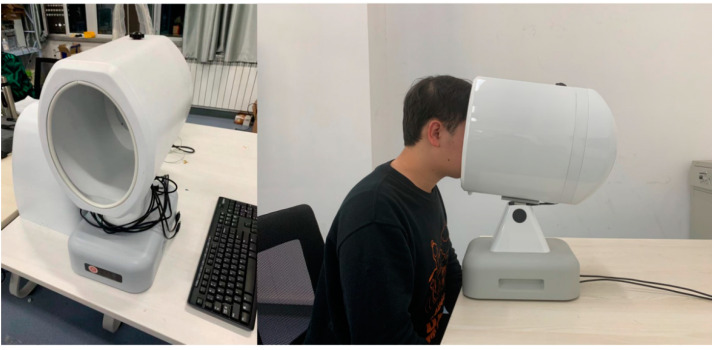
Tongue image acquisition with the tongue diagnosis instrument.

**Figure 3 micromachines-13-00501-f003:**
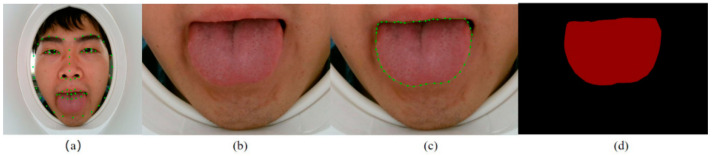
(**a**) Original image with 68 facial landmarks. (**b**) Tongue region image. (**c**) Edge annotation of tongue. (**d**) Tongue contour image.

**Figure 4 micromachines-13-00501-f004:**
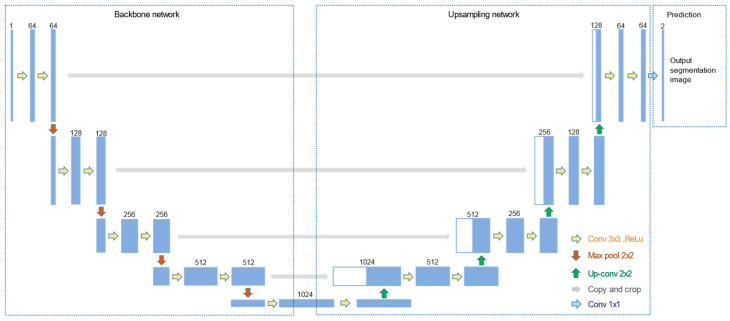
Structure of UNET.

**Figure 5 micromachines-13-00501-f005:**
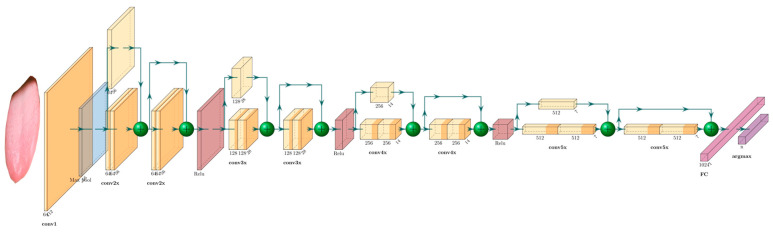
ResNet-34 Structure.

**Figure 6 micromachines-13-00501-f006:**
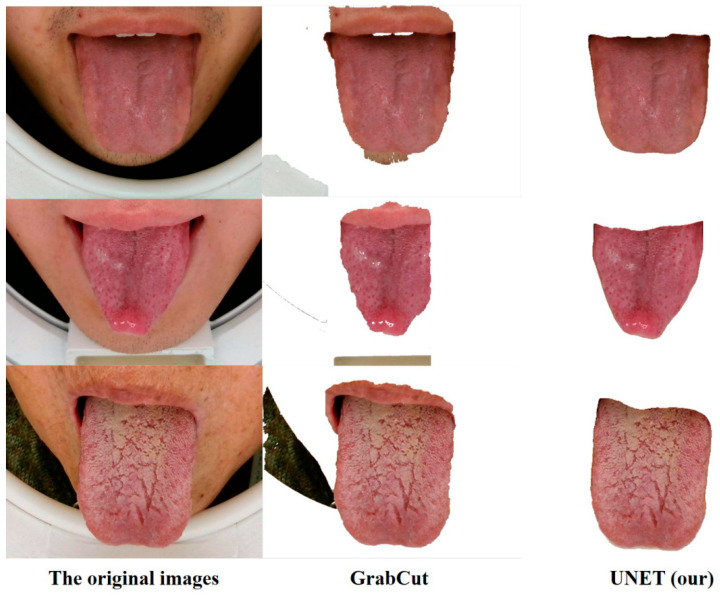
Segmentation effect comparison.

**Figure 7 micromachines-13-00501-f007:**
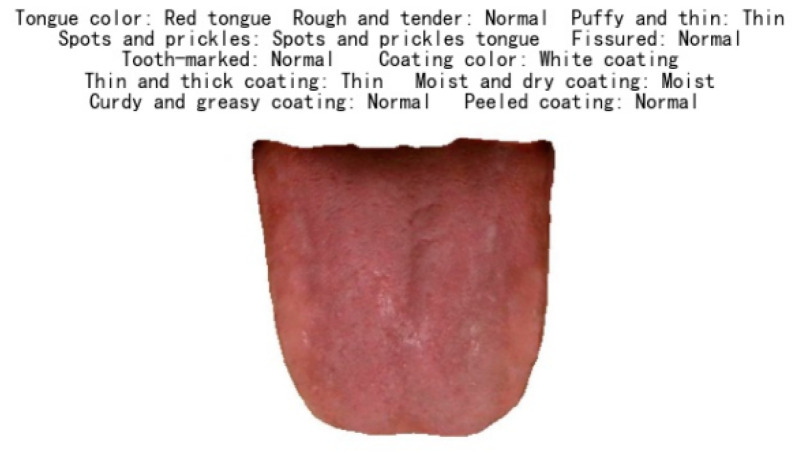
Tongue feature recognition results.

**Figure 8 micromachines-13-00501-f008:**
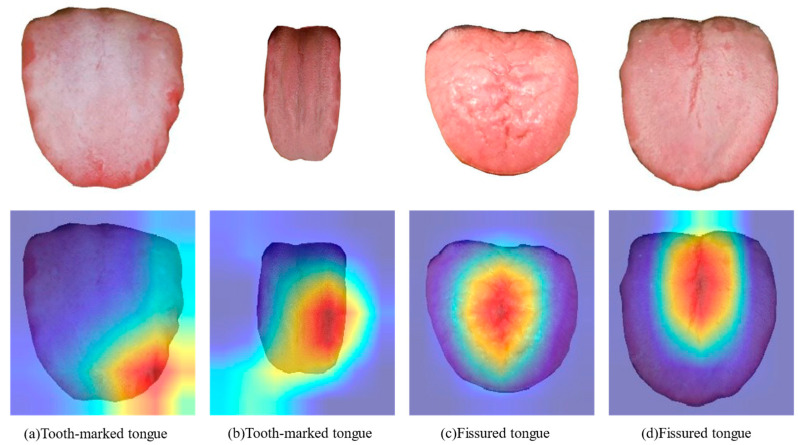
Heat map for different feature recognition.

**Table 1 micromachines-13-00501-t001:** Tongue Features.

Features	Inner-Class
Tongue color	Pale tongue, Light red tongue, Light cyanosed tongue, Red tongue, Deep red tongue, Cyanosed tongue, Ashen tongue, Red tongue borders and tip
Rough and tender tongue	Normal, Rough tongue, Tender tongue
Puffy and thin tongue	Normal, Puffy tongue, Swollen tongue, Thin tongue
Spots and prickles tongue	Normal, Spots and prickles tongue
Fissured tongue	Normal, Fissured tongue
Tooth-marked tongue	Normal, Tooth-marked tongue
Tongue coating color	White coating, Yellow coating, Grayish black coating
Thin and thick coating	Thin coating, Thick coating
Moist and dry coating	Moist coating, Slippery coating, Dry coating
Curdy and greasy coating	Normal, Greasy coating, Curdy coating
Peeled coating	Normal, Peeled coating

**Table 2 micromachines-13-00501-t002:** GrabCut and UNET segmentation results.

Method	PA	MIoU
GrabCut	79.96%	66.26%
UNET	98.54%	97.14%

**Table 3 micromachines-13-00501-t003:** Tongue feature recognition result parameters.

Feature	Acc	F1-Score
Tongue color	62.4%	55.2%
Rough and tender tongue	91.6%	83.6%
Puffy and thin tongue	86.3%	74.4%
Spots and prickles tongue	83.3%	76.5%
Fissured tongue	87.5%	82.9%
Tooth-marked tongue	86.7%	84.0%
Tongue coating color	87.5%	86.5%
Thin and thick coating	89.5%	89.2%
Moist and dry coating	87.4%	67.0%
Curdy and greasy coating	86.3%	87.2%
Peeled coating	98.9%	94.2%

## Data Availability

The data presented in this study are available on request from the corresponding author. The data are not publicly available due to this data being supplied by Key Laboratory of Instrumentation Science & Dynamic Measurement (North University of China), Ministry of Education, and so cannot be made freely available.

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
