# Peer review of "Automatic Classification Framework of Tongue Feature Based on Convolutional Neural Networks"

_micromachines, 2022, doi:10.3390/mi13040501_

Round 1

Reviewer 1 Report

The paper is focused on Automatic Classification of Tongue Feature Based on Convolutional Neural Networks to reduce the differences in diagnoses among Traditional Chinese Medicine practitioners.

The disadvantage of such study is that it has to highline performance, completeness, robustness, applicability, compared to other approaches. The current paper is well written, it has a correct flow and it proves its functionality within the proposed scenario. However discussing applicability would have increased greatly the value of the work.

The authors also should present more technical details about the experiments performed.

The methodology used is a collection of known and well-established deep learning methods. The novelty of this paper must be clearly stated at the end of the first section and the contribution and innovation of this study over previous studyies must be defined. What is the knowledge gap bridged by this study?

The related works section is unstructured . What are the main challenges in this domain? What are the limitations of the previous works, which motivate the current study? Improve the overview of state-of-the-art by using the most recent sources.

Explain, number of sample  the training set, validation set and test set of the dataset divide procedure you have used.

In Conclusions section you should discuss the limitations of your approach, if any.

Author Response

请参阅附件。

Reviewer 2 Report

The paper presented a study to identify the tong features used in Chinese medicine. The paper is well organized and needs some clarifications and text editing.

  1. Section 2.2 described the extraction of the tongue region. Was it done manually by selecting the landmarks on the image? How was it different from the segmentation of the tongue image using GrabCut and UNET?
  2. Line 88: Change "self-designed" to "custom designed".
  3. Line 163: "-4" should be in superscript.

Round 2

Reviewer 1 Report

Accept in present form.